# MOTION POINTNET: SOLVING DYNAMIC CAPTURE IN POINT CLOUD VIDEO HUMAN ACTION

## ABSTRACT

Motion representation plays a pivotal role in understanding video data, thereby elevating the dynamic capture to the forefront of action recognition tasks based on point cloud video. Previous works mainly compute the motion information in an unguided way, e.g. aggregate the spatial variations on adjacent point cloud frames using 4D convolutions or capture a point trajectory with kinematic computation like scene flow. However, the former fails to explicitly consider motion representation in corresponding frames, and the latter's reliance on tracking point trajectories becomes impractical in real-life applications due to the potential inter-frame migration of points. In this paper, we tackle the dynamic capture in point cloud video action by formulating it as solvable partial differential equations (PDEs) in feature space. Based on this intuitive design, we propose Motion PointNet, a novel method that improves the dynamic capture in point cloud video human action by constructing clear guidance for network learning. Motion PointNet is composed of a lightweight yet effective PointNet-like encoder and a PDEs-solving module for dynamic capture. Remarkably, our Motion PointNet, with merely **0.72 M** parameters and **0.82 G** FLOPs, achieves an impressive accuracy of **97.52 %** on the MSRAction-3D dataset, surpassing the current state-of-the-art in all aspects. The code and the trained models will be released for reproduction.

## 1 INTRODUCTION

Recognizing human actions within the context of 3D point cloud space necessitates more than just spatial perception, as the dynamic capture of point cloud video human action has acquired paramount significance and emerged as a forefront challenge in dynamic point cloud processing. Enhancing dynamic capture has intuitively emerged as a pivotal solution for advancing the performance of the 3D action recognition task (Liu et al., 2019; Fan et al., 2021b; 2022; 2021a; 2023; Zhong et al., 2022).

Early approaches primarily focused on spatial perception and encountered challenges when dealing with the intricate structure of point cloud video data. Choy et al. (2019) and Wang et al. (2020) tackle the point cloud video data with 4D voxelization, where information loss is inevitable. Recent methods circumvent this issue by directly processing the point cloud while emphasizing the capture of dynamics. Some of these methods implicitly capture the dynamic of the point cloud video by searching related points in the spatial-temporal space (Liu et al., 2019; Fan et al., 2021b;a). These methods rely on the sophisticated designed 4D convolutions and/or the transformer-based temporal perception, where high computational demands and inherent complexity are always accompanied. Other than that, these methods lack consideration of the motion tracking in correspondence frames explicitly, and thus may not well present the dynamic information of the point cloud sequence. Other methods try to explicitly capture the dynamics by computing point trajectory with kinematic computation, e.g. scene flow (Zhong et al., 2022) for better motion representations. Nonetheless, the approach proves unrealistic in real-life applications due to the potential inter-frame migration of points. Additionally, these approaches require the exact formalization of the point trajectory, which has proved to be a challenging endeavor (Wu et al., 2023; Liu et al., 2023). Beyond concerns about training costs and point-tracking problems, the aforementioned approaches lack proper guidance during the dynamic capture process. We contend that refining the focus of dynamic capture to a distinct objective, beyond relying solely on global supervision from the action labels, can yield significant benefits for the action recognition task.

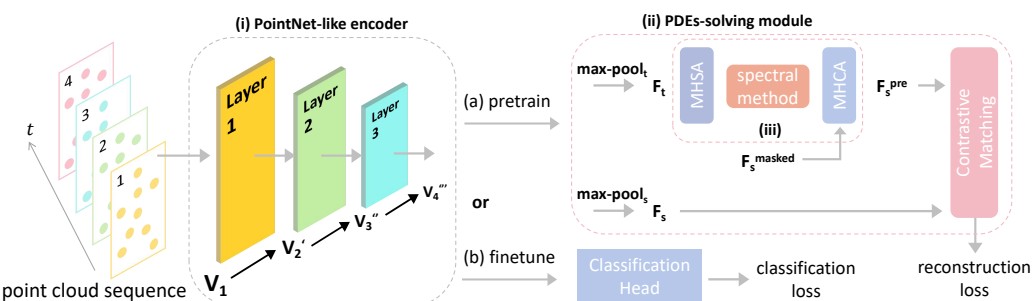

Figure 1: Overall architecture of our Motion PointNet. **PointNet-like Encoder:** Benefiting from the rolling operation, as the network goes deeper, features from the current frame are continually aggregated to the next frame, while also perceiving more spatial information with a larger spatial receptive field. **PDEs-solving module:** The network can be trained in a pretrain-finetune manner. We involve a PDEs-solving module in the pretrain stage to instruct the encoder to allocate a greater level of attention to the temporal dimension. In the finetune stage, we load the pretrained weight of the encoder only and add a classification head for supervised learning.

In light of the aforementioned considerations, we propose a new perspective by viewing dynamic capture as solvable high-dimensional partial differential equations (PDEs) in the feature space. By building a proper input-output mapping *i.e.* solving PDEs in the feature space, we can provide clear temporal guidance for the dynamic capture process. This concept is built on two intuitive facts: **(i)** PDEs in real-world applications are usually adopted in the discretized high-dimensional coordinate space, e.g. point cloud (Wu et al., 2023). **(ii)** Solving PDEs with proper input-output mapping facilitates the modeling of sequence data (Tran et al., 2021; Liu et al., 2023). Notably, our work represents a pioneering effort in applying PDEs-solving to point cloud video data. Despite challenges, we demonstrate how the concept of solving high-dimensional PDEs can address the shortcomings of other methods and enhance dynamic capture within point cloud video data in two aspects: **(i)** For the absence of proper motion tracking and temporal guidance problem, adopting PDEs-solving can explicitly capture the motion information in a well-guided way by establishing a suitable input-output mapping in a PDEs problem, instead of relying solely on the action ground truth. **(ii)** For the point tracking difficulties, we avoid the explicit kinematic equation by solving PDEs in high-dimensional space with deep models. This allows us to formalize the input-output mapping as learning operators, which can be universally approximated by model optimization.

Building upon these observations, we introduce our Motion PointNet with the utilization of PDEs solving during dynamic capture for the point cloud video action recognition task. Specifically, we pretrain the feature encoder of our Motion PointNet with a distinctive **PDEs-solving module** to improve the dynamic capture process. Our Motion PointNet contains two parts: Firstly, a **PointNet-like encoder** is adopted for spatial-temporal feature generation (see Fig.1 (i)). We adopted the set abstraction (Qi et al., 2017) on the adjacent point cloud frames to generate local variations between frames. Diverging from the 4D convolutions that process the point cloud sequence recursively and thus lead to inherent complexity, we treat all frames into a batch with a reshape operation that aligns with previous video networks (Wang et al., 2019; Lin et al., 2019). In doing so, we establish a streamlined and lightweight yet spatially and temporally conscious encoder. Secondly, an **PDEs-solving module** is applied to solve the PDEs problem in the context of dynamic capture (see Fig.1 (ii)). Given the spatial-temporal feature generated from the proposed encoder, we design our PDEs-solving as a feature reconstruction process from the masked token. Previous reconstruction-based pretrain approaches mainly address the inner data distribution patterns within spatial (Pang et al., 2022) or temporal (Wang et al., 2021) dimensions solely. Distinctively, we decouple the spatial-temporal feature and propose the reconstruction of the spatial feature from the temporal dimension. This approach prioritizes the optimization within the temporal-to-spatial Banach spaces and allows for the synthesis of spatial and temporal feature spaces during the dynamic capture process. Then we apply a classical spectral PDEs-solving method (Gottlieb & Orszag, 1977) for the mapping from temporal space to spatial space. We further adopt two attention layers for the feature aliment in different spaces (see Fig.1 (iii)).

The proposed PDEs-solving module improves the dynamic capture problem by a directive temporal-to-spatial mapping. Facilitated by the strong temporal perceptual ability of the proposed PDEs-

solving module, our Motion PointNet improves the performance of the point cloud video action recognition task by a clear margin. Extensive experiments on various benchmarks including MSRAction-3D (Li et al., 2010), NTU RGB+D (Shahroudy et al., 2016), and UTD-MHAD (Chen et al., 2015) prove the superiority of our proposal. Prominently, with only 0.72M parameters and 0.82G FLOPs, our Motion PointNet achieves an accuracy of 97.52% on the MSRAction-3D dataset.

In summary, the contributions of this work are summarized as follows: **(i)** We propose a brand-new perspective that views the dynamic capture process - a nonlinear modeling problem, as a PDEs-solving problem. The novelty lies in the reconstruction of spatial information from the temporal dimension as the PDEs-solving target. By doing so, we can establish a synthesis of spatial spaces and temporal spaces, thereby enhancing the dynamic capture process. **(ii)** We propose a lightweight PointNet-like encoder and a PDEs-solving module to enhance dynamic capture in the point cloud video action recognition task. These components form the foundation of our Motion PointNet framework, tailored for the task of point cloud video action recognition.

## 2 RELATED WORKS

### 2.1 POINT CLOUD VIDEO ACTION RECOGNITION

Point cloud video contains complex spatial-temporal information and combines an intricate structure with both unordered (intra-frame) and ordered (inter-frame) nature. Early methods either simplify its structure by dimensionality reduction using projections (Luo et al., 2018), or adopt voxelization to construct a regulated grid-based data (Choy et al., 2019; Wang et al., 2020). Similar to projections/voxel-based methods in static point clouds, those methods also faced information loss and issues with processing efficiency. Recent methods (Liu et al., 2019; Min et al., 2020; Fan et al., 2021b;a) inclined to process the point cloud video directly with PointNet-like set abstraction (Qi et al., 2017). For instance, Fan et al. (2021b) proposed a 4D convolution that implicitly captures the dynamic of adjacent point cloud frames by adopting the PointNet-like set abstraction between them and processes the point cloud sequence recursively. After that, an improved version (Fan et al., 2022) proposed to enhance dynamic capture with an additional temporal convolution. These point-based methods focus more on the motion representation and try to improve the dynamic capture process in different aspects. P4Transformer (Fan et al., 2021a) and PST-Transformer (Fan et al., 2023) captured dynamic by searching related points in the spatial-temporal space with attention-based networks. Kinet (Zhong et al., 2022) proposed a kinematics-inspired neural network and solved the dynamic capture in point cloud sequence using scene flow. Different from all the aforementioned methods, our Motion PointNet is built upon a brand-new perspective that treats dynamic capture in point cloud sequence as a solvable PDEs problem.

### 2.2 PDEs-SOLVING WITH DEEP MODELS

Our work is also related to solving PDEs numerically with deep models. The PSEs-solving problem has been widely explored with spectral methods since the last century (Gottlieb & Orszag, 1977; Fornberg, 1998). Recently, some research work explored the deep models for PDEs due to their great nonlinear modeling capability (Li et al., 2020; Tran et al., 2021; Fanaskov & Oseledets, 2022; Liu et al., 2023). In this paper, we aim to adopt the PSEs-solving in the point cloud video action recognition task. Specifically, we design a unique temporal-to-spatial mapping with a reconstruction target to enhance the dynamic capture process in the point cloud video action recognition task. To the best of our knowledge, we are the first work that adopts PDEs-solving in this task.

## 3 PROPOSED METHOD

We illustrate the proposed Motion PointNet in detail in the following sections. Fig.1 shows the overall architecture of the Motion PointNet, which is composed of a PointNet-like encoder and a PDEs-solving module. The network is trained in a pretrain-finetune manner. In the pretrain stage (see Fig.1 (a) process), we train our encoder with the PDEs-solving module under the supervision of the reconstruction loss. In the finetune stage (see Fig.1 (b) process), we load the pretrained weight of the encoder only and add a classification head for the action recognition task.

## 3.1 POINTNET-LIKE ENCODER

We design a lightweight yet effective encoder for the temporal feature generation in our Motion PointNet. Given a point cloud with $N$ points that presented as $P = \{p_1, p_2, ..., p_N\}$, where each point $p_i \in \mathbb{R}^3$ is specified by the geometric coordinates $\{x, y, z\}$. A point cloud video contains $T$ frames of point clouds presented as $V = \{P_1, P_2, ..., P_T\}$, combining characteristics of both unordered intra-frame and ordered inter-frame. Previous methods (Fan et al., 2021b;a; 2023) process the point cloud video $V$ recursively by sophisticated designed 4D convolutions. Differently, we process $T$ frames of data as batches following previous video networks (Lin et al., 2019; Wang et al., 2019), thus processing the point cloud video with the shape of $\{B \times T, N, C\}$. Here a 'batch' represents a group of data entered into the network for training, and the 'batch size' represents the number of training samples in each batch. Here the $batch\ size = B \times T$. Usually, $C = 3$ represents the spatial coordinates $\{x, y, z\}$.

We then extend the spatial encoder from PointNet++ (Qi et al., 2017) for static point clouds only to the temporal domain. For a static point cloud $P$, Qi et al. (2017) adopt a multilayer perceptron (MLP) for spatial set abstraction:

$$Feature = f(P, P^{'}), \tag{1}$$

where $f$ represents a standard PointNet++ layer. We omit some basic operations in point cloud processing to simplify the description (*e.g.* sampling and grouping for the generation of $P^{'}$ from $P$). Here $P$ functions as the support points and $P^{'}$ is the downsampled query points from the original $P$. We recommend referring to PointNet++ (Qi et al., 2017) for more details. Eq.1 can be extended and further formed as follows when using batch processing, thus easily fitting our reshaped point cloud video:

$$Feature = f(\{P_1, P_2, ..., \}, \{P_1, P_2, ..., \}^{'}) = f(V, V^{'}). \tag{2}$$

However, the simple spatial encoder still lacks temporal consciousness and cannot well present the motion information. We solve this problem by adding a *rolling* operation *i.e.* $torch.roll()$ on the temporal dimension which leads to frame misalignment in point cloud videos. In other words, we generate the support points and the query points from different point cloud frames instead of the same one to aggregate temporal features from $P_t \rightarrow P_{t+1}$:

$$Feature = f(P_t, P^{'}_{t+1}). \tag{3}$$

Here, the $P_t$ functions as the support points and $P^{'}_{t+1}$ is the downsampled query points from the next point cloud frame $P_{t+1}$. When using batch processing, Eq.3 can be reformulated as follows:

$$Feature = f(\{P_1, P_2, ..., P_t, ...\}, \{P_2, P_3, ..., P_{t+1}, ...\}^{'}) = f(V_1, V^{'}_2), \tag{4}$$

where the index of $V$ represents the temporal index of the first point cloud frame in the point cloud sequence. In this way, the spatial set abstraction is extended to the temporal domain by operating on the adjacent frames while keeping its lightness and simplicity. We naturally stack multiple PointNet++ layers to build our PointNet-like encoder. As the network delves deeper, simultaneous temporal rolling and spatial abstraction persist, resulting in the expansion of the encoder's receptive fields in both spatial and temporal dimensions. Taking a 3-layer depth encoder as an example:

$$Layer_1 = f(V_1, V^{'}_2) \quad Layer_2 = f(V^{'}_2, V^{''}_3) \quad Layer_3 = f(V^{''}_3, V^{'''}_4), \tag{5}$$

where more superscript $'$ represents the larger spatial sampling scale than the previous layer. Different from previous methods, our encoder maintains the sequence length $T$ while aggregating temporal information from the current frame to the next frame, greatly enhancing the local information density of our features.

## 3.2 PDEs-SOLVING MODULE

We introduce a PDEs-solving module that intensifies the network's focus on dynamic capture. Traditional methods relying solely on global supervision from the action labels lack temporal guidance, potentially leading to an inadequate representation of dynamics in the point cloud video action recognition. To address this limitation, the proposed PDEs-solving module decouples the spatial-temporal feature and builds a PDEs problem by a directive temporal-to-spatial mapping. Together with the contrastive loss (presented later) in the pretrain stage, our PDEs-solving module refines the focus of dynamic capture to a distinct objective. By doing so, we expect to provide temporal guidance.

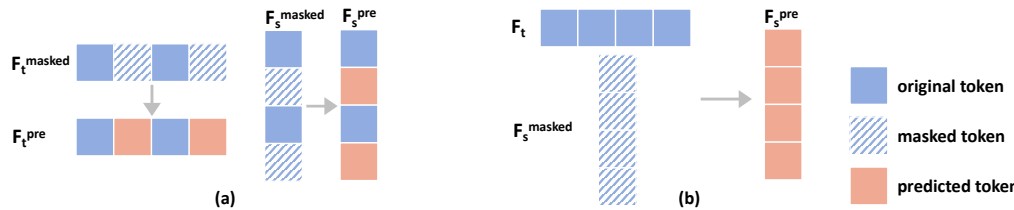

Figure 2: Comparison between reconstruction target for (a) inner data distribution and (b) dynamic capture in temporal feature.

### 3.2.1 BUILDING TEMPORAL-TO-SPATIAL MAPPING

Given the spatial-temporal feature acquired from the encoder $Feature \in \mathbb{R}^{T \times M}$, where $M < N$ is the number of spatial regions after aggregation. Subsequently, we proceed with max-pooling in both the temporal and spatial dimensions, respectively, and obtain the sub-global temporal feature $F_t \in \mathbb{R}^T$ and the sub-global spatial feature $F_s \in \mathbb{R}^M$. Existing reconstruction-based self-supervised learning mainly addresses the inner data distribution patterns by reconstructing masked tokens (see Fig.2 (a)). These methods focus on either the spatial or temporal dimension solely and are usually sensitive to mask ratio. Furthermore, simply recovering the masked data does not meet our purposes of dynamic capture. Distinctively, we target reconstructing the $F_s$ from a learnable parameters set $F_s^{masked}$ by reversing the $F_t$ with spectral methods (see Fig.2 (b)). Both the $F_t$ and the $F_s$ are in the Banach spaces $\mathcal{F} = \mathcal{F}(\mathcal{D}; \mathbb{R}^{d_F})$, where $\mathcal{D} \subset \mathbb{R}^d$ is a bounded open set. Based on assumptions from Lu et al. (2021) and Li et al. (2021b), we can solve the PDEs with a deep model $\mathcal{M}_\theta$ by approximating the optimal operator. This process can be formulated as follows:

$$\mathcal{M} : F_t \xrightarrow{\theta} F_s, \tag{6}$$

where $\theta$ is the learnable parameter set. Our PDEs-solving module directly incorporates the feature variations over time as the variable $\theta$ in the modeling process. This allows a more accurate and nuanced representation of how point cloud features evolve over time compared to previous temporal modeling approaches.

### 3.2.2 SOLVING PDEs MAPPING

We then introduce the core of the PDEs-solving module with a spectral method. The approximating of $\theta$ in Eq.6 can be formulated as follows:

$$\mathcal{M}_\theta = \sum_{i=1}^{O} w_i \mathcal{M}_{\theta,i}, \tag{7}$$

where $O$ is the number of operators and $w_i$ is learnable weight. As shown in Fig.1 (iii), the core of the PDEs-solving module is composed of a multi-head self-attention (MHSA) layer, a spectral method layer, and a multi-head cross-attention (MHCA) layer. Although a simplistic deep model (Liu et al., 2023) can be used for PDEs-solving, the design attempts to learn the operator as a whole ($O = 1$) while still maintaining network efficiency is challenging due to the complexities of input-output mappings in high-dimensional space (Wu et al., 2023; Karniadakis et al., 2021). We tackle this problem with the combination of the attention mechanism and the classic spectral method (Tolstov, 2012) for PDE, which decomposes complex nonlinear mappings into multiple basis operators, while also holding the universal approximation capacity with theoretical guarantees.

For every $\mathbf{f_t} \in F_t$, we use trigonometric as the basis operators following Lu et al. (2021) and Li et al. (2021b):

$$\mathcal{M}_{\theta,(2k-1)}(\mathbf{f_t}) = sin(k\mathbf{f_t}), \ \mathcal{M}_{\theta,(2k)}(\mathbf{f_t}) = cos(k\mathbf{f_t}), \ k \in \{1, ..., N/2\} \tag{8}$$

where $N$ is even. Then, the calculation of the mapping output can be formulated as follows:

$$F_{t \to s} = F_t + w_{sin}[\mathcal{M}_{\theta,(2k-1)}(\mathbf{F_t})]_{k=1}^{O/2} + w_{cos}[\mathcal{M}_{\theta,(2k)}(\mathbf{F_t})]_{k=1}^{O/2}. \tag{9}$$

Building upon the spectral method we elaborate above, we use a standard MHSA (Vaswani et al., 2017) layer to enhance the temporal feature before feeding into the spectral method layer. Furthermore, we align the two different Banach spaces of the mapping output $F_{t \to s}$ and $F_s^{masked}$ by an MHCA layer and output the predicted $\hat{F}_s$. [1] The $F_s^{masked}$ is initialized with the same shape as $F_s$.

---

[1]Due to page limitations in the main text, we further detail this process in the Appendix A.

### 3.2.3 Contrastive Matching Loss

We compare the predicted $\hat{F}_s$ with the ground truth $F_s$ using a contrastive-based InfoNCE loss (Oord et al., 2018) in our pretrain stage to refine the focus of dynamic capture to a distinct objective. We consider both the input $F_t$ and the output $F_s$ to contain more abstract information about the spatio-temporal features. By treating the $\mathbf{f}_-$ in $Feature \in \mathbb{R}^{T \times M}$ before applying spatial/temporal pooling as the negative sample, we force the model to learn the implicit mapping between the spatial and temporal space, instead of reconstructing the de-pooling feature by closer both Ft and Fs to $Feature$ (which is against the learning objectives). For every $\hat{\mathbf{f}}_\mathbf{s} \in \hat{F}_s$, we treat the corresponding token in $F_s$ as positive sample $\mathbf{f}_{\mathbf{s}+}$. The loss function can be formulated as follows:

$$\mathcal{L} = \sum_{\hat{\mathbf{f}}_{\mathbf{s},i} \in \hat{F}_s} -log \frac{exp(\hat{\mathbf{f}}_{\mathbf{s},i}^T \mathbf{f}_{\mathbf{s}+}/\tau)}{exp(\hat{\mathbf{f}}_{\mathbf{s},i}^T \mathbf{f}_{\mathbf{s}+}/\tau) + \sum_{\mathbf{f}_{-,j} \in Feature} exp(\hat{\mathbf{f}}_{\mathbf{s},i}^T \mathbf{f}_{-,j}/\tau)}, \tag{10}$$

where $\tau$ is a temperature that controls the network sensitivity to positive and negative samples. Several ablation studies in Section 4.3 underscore the superiority of our loss design.

## 4 Experiment

### 4.1 Experimental Settings

We evaluate the proposed Motion PointNet on three benchmarks including the MSRAction-3D (Li et al., 2010) dataset, NTU RGB+D (Shahroudy et al., 2016) dataset and UTD-MHAD (Chen et al., 2015) dataset for point cloud video action recognition. We conduct all our experiments on the NVIDIA A100 GPUs. Following the previous works (Zhong et al., 2022; Fan et al., 2021b; 2023; Chen et al., 2015), we use default data splits in all evaluated datasets for fair comparisons. In most of the experiments, we pretrain our Motion PointNet with the PDEs-solving module on the training data split with a 24-frame video input and finetune the encoder we proposed in Sec.3.1 with a classification head. We report our Motion PointNet results from the finetuned models.

### 4.2 Action Recognition Results

**MSRAction-3D** dataset includes 567 depth map sequences of 20 action classes performed by 10 subjects. To generate point cloud videos from the original data, we adopt the standard method following Liu et al. (2019) and Fan et al. (2021b;a), and report the average accuracy of our experiment over 10 runs following the convention. We compare our Motion PointNet to prior works in Tab.1. Our report proves that the proposed method outperforms the current SOTA by significant margins, gaining a **+3.79%** accuracy with 24-frame input. Furthermore, it maintains superior performance with reduced frame input (12/16-frame), demonstrating the robustness of the proposed Motion PointNet. The performance on 4/8-frame MSRAction-3D indicates a slight limitation on short video input with a comparable accuracy. Indicate that the simple and explicit temporal informativeness of adjacent frames is proportional to the length of the input video.

Notably, the proposed method not only attains state-of-the-art performance but also surpasses existing models in terms of model parameters, complexity, and running time. As illustrated in Tab.2, previous approaches that rely on sophisticated 4D convolutions are usually accompanied by intricate computational demands and substantial learning parameters. For instance, PSTNet (Fan et al., 2021b) is composed of a hierarchical architecture with 4-layer 4D convolutions, which result in over 50G FLOPs. When further improving the performance by even more complex networks (Fan et al., 2021a; 2023), the learning parameters reach a staggering 40M+. Differently, our Motion PointNet miniaturizes the model in both FLOPs and learning parameters to the SOTA level. Remarkably, our Motion PointNet surpasses the current state-of-the-art with a **0.55**G FLOPs (when comparing at a 16-frame input) and **0.72**M learning parameters. We further visualize the points corresponding to high feature response in Fig.4. As we can see, the main moving part of actions (*e.g.* swinging arms in the golf swing) are highlighted, which is consistent with the proposed intuition.

**NTU RGB+D** dataset contains 60 action classes and 56,880 video samples, which is a large-scale dataset consisting of complex scenes with noisy background points. We report the results of the cross-subject and cross-view scenarios following the official data partition (Shahroudy et al., 2016).

Table 1: Comparison with current state-of-the-art on MSRAction-3D dataset.

| Methods | Accuracy(%) of different frame rate | | | | |
|---|---|---|---|---|---|
| | 4-frame | 8-frame | 12-frame | 16-frame | 24-frame |
| MeteorNet (Liu et al., 2019) | 78.11 | 81.14 | 86.53 | 88.21 | 88.50 |
| P4Transformer (Fan et al., 2021a) | 80.13 | 83.17 | 87.54 | 89.56 | 90.94 |
| PSTNet (Fan et al., 2021b) | 81.14 | 83.50 | 87.88 | 89.90 | 91.20 |
| SequentialPointNet (Li et al., 2021a) | 77.66 | 86.45 | 88.64 | 89.56 | 91.94 |
| PointMapNet (Li et al., 2023) | 79.04 | 84.93 | 87.13 | 89.71 | 91.91 |
| PSTNet++ (Fan et al., 2022) | 81.53 | 83.50 | 88.15 | 90.24 | 92.68 |
| Kinet (Zhong et al., 2022) | 79.80 | 83.84 | 88.53 | 91.92 | 93.27 |
| 3DInAction (Ben-Shabat et al., 2023) | 80.47 | 86.20 | 88.22 | 90.57 | 92.23 |
| PST-Transformer (Fan et al., 2023) | 81.14 | 83.97 | 88.15 | 91.98 | 93.73 |
| **Motion PointNet** | 79.46 | 85.88 | 90.57 | 93.33 | **97.52** |

Table 2: Qualitative results for efficiency evaluation on MSRAction-3D. Notice that the reported runtime results are on 24-frame MSRAction-3D.

| Methods | flames | FLOPs(G) | Param.(M) | Acc.(%) | time(ms) |
|---|---|---|---|---|---|
| PSTNet (Fan et al., 2021b) | | 54.09 | 8.44 | 89.90 | 63.88 |
| MeteorNet (Liu et al., 2019) | | 1.70 | 17.60 | 88.21 | 80.11 |
| P4Transformer (Fan et al., 2021a) | 16 | 40.38 | 42.07 | 89.56 | 25.18 |
| PST-Transformer (Fan et al., 2023) | | - | 44.20 | 91.98 | 69.37 |
| Kinet (Zhong et al., 2022) | | 10.35 | 3.20 | 91.92 | - |
| **Motion PointNet** | 16/24 | **0.55**/0.82 | **0.72** | 93.33/**97.52** | **1.17** |

We compare our Motion PointNet to prior works in Tab.3. Our report proves that the proposed method maintains its superiority on the large-scale dataset. Our Motion PointNet is superior to most of the methods with different input modalities including depth map, skeleton, and dense points. It consistently outperforms the large model including PSTNet (Fan et al., 2021b), P4Transformer (Fan et al., 2021a), and PST-Transformer (Fan et al., 2023) in both the cross-subject (92.9% accuracy) and cross-view (98.0% accuracy) protocols, while forming a way more lightweight network. As shown in Tab.4, our Motion PointNet offers consistent superiority in lightweight regarding model parameters (1.64M parameters) and computational complexity (15.47G FLOPs).

We also report the hyper-settings of our Motion PointNet for the two aforementioned datasets in Tab.5. We modified the basis hyperparameters in the table and selected the best-setting group in our experiments. The NTU RGB+D dataset requires a deeper network due to its complex scenes and large scale. Notice that our PDEs-solving module is also lightweight with only additional +5.2M parameters and +0.2G FLOPs in both settings.

**UTD-MHAD** dataset contains 27 classes and 861 data sequences for action recognition. We apply our Motion PointNet to the UTD-MHAD benchmark and compare the proposed approach with current SOTA methods. The encoder settings of the Motion PointNet are consistent with the settings for the NTU RGB+D benchmark. Results in Tab.6 illustrate the accuracy of different approaches. Our Motion PointNet maintains its superior performance with the highest accuracy of 92.79%.

### 4.3 ABLATION STUDIES

We conduct extensive ablation experiments on the proposed Motion PointNet. Results in Tab.7 show that the proposed encoder itself has outperformed the PST-Transformer (Fan et al., 2023) with a 95.76% accuracy. Furthermore, the PDEs-solving module brings a significant improvement (+1.75% accuracy) to our encoder. We also implement our PDEs-solving module on the PST-Transformer. After pertaining together with our PDEs-solving module, the PST-Transformer encoder also achieved a +1.32% accuracy improvement. This underscores the universality of our PDEs-solving module and the applicability of the PDE-solving perspective across different scenarios in point cloud video action recognition.

We further validate the effectiveness of different components in the PDEs-solving module. Results are shown in Tab.8. Firstly, we assess the individual contributions of the three layers that constitute

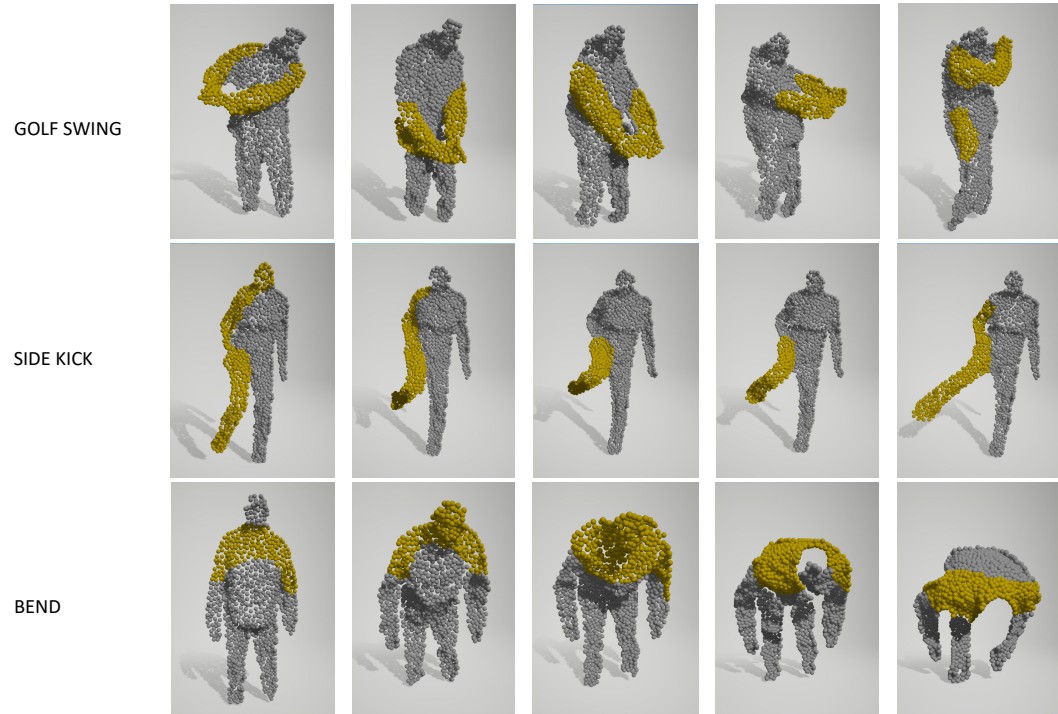

Figure 3: Visualization of high feature response on MSRAction-3D dataset. High response points are marked in orange, which are selected based on the magnitude of the feature response. We choose binary representation for clarity in visualization.

Table 3: Compare to the current state-of-the-art on NTU RGB+D dataset.

| Methods | Modalities | Cross Subject | Cross View |
|---|---|---|---|
| Li et al. (Li et al., 2018) | | 68.1 | 83.4 |
| Wang et al. (Wang et al., 2018) | depth map | 87.1 | 84.2 |
| MVDI (Xiao et al., 2019) | | 84.6 | 87.3 |
| SkeleMotion (Caetano et al., 2019) | | 69.6 | 80.1 |
| DGNN (Shi et al., 2019) | skeleton | 89.9 | 96.1 |
| MS-G3D (Liu et al., 2020) | | 91.5 | 96.2 |
| 3DV (Wang et al., 2020) | | 88.8 | 96.3 |
| P4Transformer (Fan et al., 2021a) | | 90.2 | 96.4 |
| PST-Transformer (Fan et al., 2023) | | 91.0 | 96.4 |
| Kinet (Zhong et al., 2022) | | 92.3 | 96.4 |
| PSTNet (Fan et al., 2021b) | points | 90.5 | 96.5 |
| PSTNet++ (Fan et al., 2022) | | 91.4 | 96.7 |
| PointMapNet Li et al. (2023) | | 89.4 | 96.7 |
| SequentialPointNet (Li et al., 2021a) | | 90.3 | 97.6 |
| **Motion PointNet** | points | **92.9** | **98.0** |

Table 4: Qualitative results for efficiency evaluation on NTU RGB+D dataset.

| Model | FLOPs(G) | PARAMS(M) |
|---|---|---|
| PSTNet (Fan et al., 2021b) | 19.58 | 8.52 |
| PointMapNet Li et al. (2023) | - | 2.65 |
| P4Transformer (Fan et al., 2021a) | 48.63 | 65.17 |
| PST-Transformer (Fan et al., 2023) | 48.68 | 65.19 |
| GeometryMotion-Net (Liu & Xu, 2021) | 68.42 | 40.44 |
| **Motion PointNet** | **15.47** | **1.64** |

Table 5: Hyper-settings of our Motion PointNet.

| | Dataset | MSRAction-3D | NTU RGB+D |
|---|---|---|---|
| Encoder Setting | Input points × frames | 2048 × 24 | 2048 × 24 |
| | Number of encoder layers | 3 | 5 |
| | Spatial stride | 32, 8, 2 | 8, 8, 1, 1, 4 |
| | K-neighbors | 48, 32, 8 | 32, 48, 16, 24, 32 |
| | Output feature channel | 1024 | 1024 |
| PARAMS(M) | w/o PDEs-solving | 0.72 | 1.64 |
| | w/ PDEs-solving | 5.95 | 6.83 |
| FLOPs(G) | w/o PDEs-solving | 0.82 | 15.47 |
| | w/ PDEs-solving | 1.06 | 15.73 |

Table 6: Compare to the current state-of-the-art on UTD-MHAD benchmark.

| Methods | Accuracy(%) |
|---|---|
| SequentialPointNet (Li et al., 2021a) | 92.31 |
| PointMapNet (Li et al., 2023) | 91.61 |
| Motion PointNet | 92.79 |

Table 7: Ablation on PDEs-solving module. Experiments are conducted on the MSRAction-3D benchmark.

| Methods | Accuracy(%) |
|---|---|
| PST-Transformer (Fan et al., 2023) | 93.73 |
| + PDEs-solving | 95.05 (**+1.32**) |
| Our Encoder | 95.76 |
| + PDEs-solving | 97.52 (**+1.76**) |

the core of PDEs-solving module. We observe that the spectral method primarily contributes to the performance enhancement, with the MHSA and MHCA layers also demonstrating their indispensability. The following results prove the utility of our contrastive matching loss. Other measures including cosine similarity and L2 distance are not ideal when we want to maximize the similarity between representations since they are either insensitive to linear scale or unbounded and harder to optimize. Finally, we validate different reconstruction targets in the PDEs-solving module. The performance decays when we attempt to capture the dynamics by solving $F_s \rightarrow F_t$ instead of $F_t \rightarrow F_s$. We hypothesize that this phenomenon arises because $F_t$ preserves a higher degree of integrated temporal information during the process of solving PDEs.

Table 8: Ablation of different components of PDEs-solving module. Experiments are conducted on the MSRAction-3D benchmark.

| Settings | | Accuracy(%) |
|---|---|---|
| full PDEs-solving module | | 97.52 |
| PDEs-solving core | w/o MHSA | − 0.71 |
| | w/o spectral method | − 1.14 |
| | w/o MHCA | − 0.41 |
| w/o contrastive matching | L2 similarity | − 0.63 |
| | Cosine similarity | − 0.43 |
| w/ contrastive matching | InfoNCE loss | ± 0. |
| reconstruction targets | $F_s \rightarrow F_t$ | − 0.68 |
| | $F_t \rightarrow F_s$ | ± 0. |

## 5 CONCLUSION

We have presented a novel architecture called Motion PointNet. Firstly, we proposed to view the dynamic capture process in the point cloud video action recognition task as a PDEs-solving problem. Based on this perspective, we designed a PDEs-solving module and a lightweight PointNet-like encoder to construct our Motion PointNet. The proposed method refines the focus of dynamic capture to a distinct objective with clear temporal guidance. Thus bringing significant improvement to the three evaluated benchmarks. The extensive experimental results also supported our superiority in versatility and model miniaturization. In future work, we aim to further extend our Motion PointNet to an extensive range of point cloud video understanding tasks including segmentation, detection, and object tracking.

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

## A  DETAILS FOR THE MHSA AND MHCA LAYERS

The MHSA layer and MHCA layer share the same structure but with different inputs. The MHSA layer uses the same $F_t$ for query, key, and value generation. The MHCA layer uses the $F_s^{masked}$ for query generation, and the mapping output $F_{t \to s}$ for key and value generation.

**MHSA**  Firstly, the $F_t$ is fed into a standard MHSA (Vaswani et al., 2017) module. We add the frame indexes as the position embedding. This process can be formulated as follows:

$$F_t = \text{PE}([1, 2, ..., T]) + F_t, \tag{11}$$

$$H_m = \text{Softmax}(\{F_t W_m^Q\} \times \{F_t W_m^K\}^{Transpose}/\sqrt{d}) \times F_t W_m^V, \tag{12}$$

$$F_t = \text{Concat}(H_1, \ldots, H_m), \tag{13}$$

where $\text{PE}(\cdot)$ is the positional encoding function that embeds the frame index to high-dimension. $W_m^Q, W_m^K, W_m^V$ are learnable weights of the $m$th head for query, key, and value respectively. And $d$ is the number of feature channels.

**MHCA**  After getting the mapping output $F_{t \to s}$ from our spectral method layer, we align the two different Banach spaces of $F_{t \to s}$ and $F_s$ by an MHCA layer. We execute this process with a learnable masked parameters set that is aligned with the $F_s$. This process can be formulated as follows:

$$H_m = \text{Softmax}(\{F_s^{masked} W_m^Q\} \times \{F_{t \to s} W_m^K\}^{Transpose}/\sqrt{d}) \times F_{t \to s} W_m^V, \tag{14}$$

$$\hat{F}_s = \text{Concat}(H_1, \ldots, H_m), \tag{15}$$

the $F_s^{masked}$ is initialized with the same shape as $F_s$. We then match the predicted $\hat{F}_s$ and the ground truth $F_s$ as the supervision of the PDEs-solving.

## B ABLATION ON TRAINING SETTINGS

Our two-stage training process is designed to first allow the network to learn robust spatio-temporal features from the encoder and then to refine this representation using the classification head in the second stage. In this way, we can keep our encoder as lightweight as possible while still enforcing its strong learning ability. To further demonstrate the effectiveness of our proposed method, we add two baseline comparisons below. The experiments are conducted on the MSR-Action3D Dataset with 24 frames.

Table 9: Ablation of different training settings for the Motion PointNet. Experiments are conducted on the MSRAction-3D benchmark with 24 frames.

| Settings | Accuracy(%) |
|---|---|
| pretrain + finetune | 97.52 |
| setting 1 | − 1.77 |
| setting 2 | − 1.41 |

Here setting 1 indicates training our encoder with the classification head using the same number of iterations of two-stage training. Setting 2 indicates finetuning the classification head while freezing the encoder to evaluate the pretrained representation.

## C VISUALIZATION

We also report the feature response from PointNet++ (Qi et al., 2017) as further comparison with our Motion PointNet. The observation reveals that the PointNet++ model exhibits a response to regions where geometric features are distinctly pronounced, such as the head, shoulders, and arms, irrespective of whether these areas constitute the primary focus of the action.

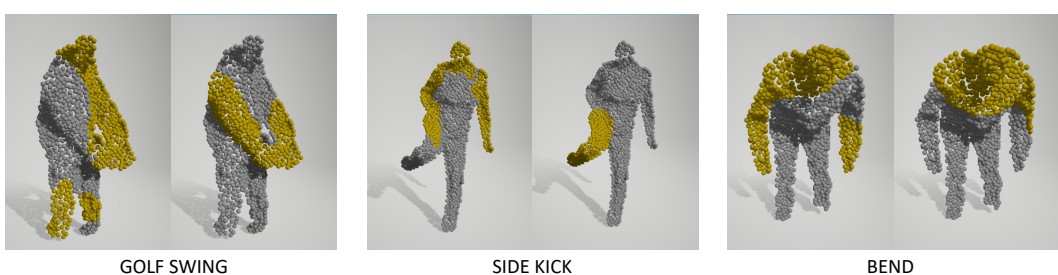

GOLF SWING  SIDE KICK  BEND

**Left:** PointNet++ **Right:** Our Motion PointNet

Figure 4: Visualization comparison between PointNet++ and our Motion PointNet. High response points are marked in orange, which are selected based on the magnitude of the feature response. We choose binary representation for clarity in visualization.

