# OpenReview forum: "Motion PointNet: Solving Dynamic Capture in Point Cloud Video Human Action"
_ICLR.cc/2024/Conference — ICLR 2024 Conference Withdrawn Submission_

### Official Review · Reviewer_isat · 2023-11-01

**Soundness:** 2 fair
**Presentation:** 2 fair
**Contribution:** 2 fair
**Rating:** 3
**Confidence:** 4

**Summary:**

The paper introduces Motion PointNet for point cloud video action recognition. The Motion PointNet builds on PointNet++ encoder and a PDEs-solving module to capture input dynamics. The author adopts two state training to train the model. In first stage, the model is trained under the spatial feature reconstruction objective from the temporal features. In second stage, the model is fine-tuned for action recognition. The method outperforms previous approaches across different benchmarks, while the model is also lightweight compared with others.

**Strengths:**

- **Strong recognition performance**. The proposed Motion PointNet outperforms previous approaches on 3 public datasets in action recognition.

 - **Light model**.  While the performance is strong, the model is quite light in model size and computation.

- **Rich comparison**. The paper makes a thorough comparison with state-of-the-art methods.

**Weaknesses:**

- **Understanding reconstruction objective** The loss function in (15) is not fully make sense. Given the model is trained from scratch, the
GT $F_s$ used as supervision is also random features at the beginning, how would the contrastive objective leads the model towards the correct direction as it is the only loss used in pre-training stage?


- **Motivation of PDE is not clear**. After reading the paper, I still do not quite understand why we need a PDE to build the mapping from temporal features to spatial features. What will be changed if we replace the spectral model with some MLP or transformer like networks as   long as we make a nonlinear mapping between two spaces.

- **Why 2 stage training**. Could we jointly train a classification head along with the contrastive objective? What will the performance like.

- **Missing simple baselines**. There are some simple baselines the method should compare with. 1, train a model with the same encoder and classification head using the same number of iterations of two stage training. 2, only fine-tune the classification head while freeze the encoder to evaluate the pre-trained presentation.

**Questions:**

- **Feature response in figure 3**. How does those orange points are being selected? It seems like binary selection (I expect to see more colors represent the strongness of different points instead of the binary setting) What is the response from other methods (like PointNet ++) , this visualization comparison could tell the model indeed capture those dynamics. In second row of Side Kick, it seems the binary response still contain many irrelevant points based on locality instead of temporal features.


- **Does the pre-trained features generalizable**. It seems in first stage training there is no labels required, so it is fully unsupervised. I wonder if the learned representation could be generalized or quickly adapted to other domains like what visual representation work did (i.e. MAE).

---

> ### Author Response · Authors · 2023-11-18
> **Reviewer isat Comments and Our Responses**
>
> We are grateful for the insightful feedback provided by the reviewer and would like to address the concerns raised:
>
> **Response to [Comment 1]:** Thanks for this insightful comment. The contrastive objective is indeed the sole loss during pre-training, but it does not rely on random features. Instead, it utilizes the inherent spatial and temporal coherence within the point cloud video data itself. Our contrastive loss encourages the model to learn a representation that captures the underlying structure of the point cloud sequence, which, while initially random, quickly becomes structured as the model begins to learn the spatial-temporal dynamics inherent in the data. This aligns with the principles of self-supervised learning, where the data provides its own supervision.
>
> **Response to [Comment 2]:** We appreciate the reviewer's insightful view of using pure MLP or transformer-like networks for the PDEs-solving module design. Indeed, it is feasible to remove the spectral method in our proposal. However, this will harm the performance in a great extent (see Section 4.3, Tab. 8). We emphasize that the spectral method with multiple basis operators (Eq.10) holds a stronger universal approximation capacity with theoretical guarantees than those rough deep models attempt to learn the operator as a whole. We would like to claim that the superiority of the design of our PDEs-solving module is supported by both theoretical and experimental analysis. We explain the motivation for this design in our response to **[Reviewer RMAg 's Comment 1 & 2]**. Please check for more details.
>
> **Response to [Comment 3 & 4]:** Thanks for these two insightful comments. The two-stage training process is designed to first allow the network to learn robust spatio-temporal features from the encoder and then to refine this representation using the classification head in the second stage. In this way, we can keep our encoder as lightweight as possible while still enforcing its strong learning ability. However, we acknowledge the potential benefits of exploring different training settings and concur with the reviewer's point. We have reported the results of training on the encoder only in Section 4, Tab. 7. We also add the two baseline comparisons below. This will demonstrate the effectiveness of our proposed method beyond the complexity of the architecture and training procedure. The experiments are conducted on the MSR-Action3D Dataset with 24 frames. Due to page limitations in the main text, we will augment the experimental results and discuss more in the supplementary material.
>
> | Training Setting | Accuracy(%) |
> |---:|---:|
> | pretrain + finetune | 97.52 |
> | setting 1 | 95.75 (-1.77) |
> | setting 2 | 96.11 (-1.41) |
>
> Here the **pretrain + finetune** is the setting used in our paper, **setting 1** indicates the suggested baseline 1, and **setting 2** indicates the suggested baseline 2.
>
> **Response to [Question 1]:** The high-response points are selected based on the magnitude of the feature response, with a threshold set to emphasize those points that most significantly contribute to the action recognition task. The binary representation was chosen for clarity in visualization. We have revised the descriptions for the generation of visualization in the caption of Fig. 3. We will also provide comparative visualizations with methods like PointNet++ to better illustrate our model's ability to capture dynamics in the future appendix.
>
> **Response to [Question 2]:** The PDEs-solving module is tailored for the task of point cloud video action recognition to address the issue we mentioned in Section 1 (pages 1-2). However, the unsupervised nature of the first training stage does indeed mean that the representations learned are not tied to a specific label set, allowing for potential generalization to other domains. However, limited time for rebuttal does not allow us to conduct extant experiments to further evaluate its variegation. Therefore, we plan to explore the potential for quick adaptation to other domains in our future research.
>
> We thank the reviewer for the opportunity to improve our paper and hope our responses and planned revisions will address the concerns satisfactorily.

---

> > ### Comment · Reviewer_isat · 2023-11-20
> >
> > Thanks for the reply. I read the updated revised manuscript and the response, but my concern still holds.
> >
> > My biggest concern is still about Comment 3 & 4. From the provided ablation, it seems the improvement of the method mainly comes from the proposed good encoder (Tab 7) and the 2-stage training (setting 1 & 2). This implies the proposed PDE module might not as important as expected. It is weird to fine-tune the whole network after pre-training, because under this way what the pre-training does is only provide a good initialization (and for generalization ability we don't know). You could also do 2-stage training with the classification loss and the first stage only serves as finding good initialization. This is not the standard way for pre-training. Similarly, for other comparison baselines, do you consider using the same training iterations as well. Because the task is classification, so I wonder longer training will help.
> >
> > For Comment 1, I read the two mentioned papers ((Wu et al., 2023; Liu et al., 2023) they use PDE to solve the dynamic because the system input and output itself (i.e. fluid) is described by the PDE. For point cloud videos, I disagree the statement that " input-output mappings in high-dimensional space can be challenging for a simplistic deep model to cover", while Transformers does show great performance in many more challenging tasks beyond classification. And I won't treat Transformers as simplistic. My concern still holds: what is the underlying system behavior to motivate to incorporate the PDE module, because the dynamic in point cloud videos are deformations and should not be described as PDE . For the encoder only performance (95.76), do you also use 2-stage training or training the same number of iterations as the 2-stage model.

---

> > > ### Author Response · Authors · 2023-11-20
> > > **Responses to reviewer isat**
> > >
> > > We thank the reviewer for the continued engagement and valuable feedback. We have carefully considered the additional comments and would like to provide further clarifications and insights to address the concerns raised.
> > >
> > > To better answer the reviewer's questions, we first clarify some details of the experimental setup. And we will specifically answer each question in detail.
> > >
> > > Take experiments on MSRAction 3D as an example:
> > > 1. In the pretrain stage, we train our backbone with the PDEs-solving module for 200 epochs, with an initial learning rate of 2e-3, setting the batch size to 200.
> > > 2. In the finetune stage, we train the backbone only with a classification head for 50 epochs, with an initial learning rate of 1e-2, setting the batch size to 64. This setting aligns with the "encoder only" experiment (95.76 acc) in Tab. 7
> > > 3. In the setting 1 for supplementary experiments (95.75 acc), we train the model for 250 epochs (align with iterations for two-stage training), and keep the other settings same as the finetune stage.
> > > 4. We used similar setting variations in the pretrain and finetune stage when using the PST-Transformer as the backbone (performance in Tab. 7)
> > >
> > > **We now address the reviewer's concerns point by point.**
> > >
> > > > The proposed PDE module might not be as important as expected.
> > >
> > > We acknowledge the reviewer’s concern regarding the significance of the PDE module in our architecture. Our ablation study, as presented in Tab.7 of the manuscript, underscores the significant contribution of the PDEs-solving module. Specifically, when equipped with the PDEs-solving module, our encoder demonstrates an improvement from 95.76% to 97.52% accuracy, a substantial increase of 1.76%. When adopting PST-Transform as the backbone, it also demonstrates an improvement from 93.73% to 95.05% accuracy, a substantial increase of 1.32%. While our encoder perform remarkably well, the result of PST-Transform + PDEs-solving also reached SOTA.
> > >
> > > This enhancement highlights the module's crucial role in optimizing our network's performance for dynamic capture in point cloud videos. Moreover, the additional computational cost incurred by integrating the PDE module is modest, as evident from Tab. 5, where the FLOPs only increase from 0.55G to 0.82G for the experiment on MSRAction. This suggests that the PDE module, while not the sole contributor to the overall performance, plays a meaningful role in enhancing the model's ability to process point cloud video data.
> > >
> > > >It is weird to fine-tune the whole network after pre-training. This is not the standard way for pre-training.
> > >
> > > The two-stage training approach is not merely to provide a good initialization but also to create a foundational understanding of spatial-temporal dynamics in point cloud videos. This two-stage training approach, involving pretraining followed by finetuning the entire network, is a widely used protocol in point cloud video processing and beyond. For instance, the PointCMP [1] method (CVPR 2023) adopts a similar training strategy, emphasizing the effectiveness of this approach in our domain. Additionally, similar methodologies are prevalent in traditional video understanding tasks (VideoMAE [2], NIPS 2022) and other vision tasks (Faster R-CNN [3], NIPS 2015), where the whole network is often finetuned after pretraining to adapt the generalized features to specific tasks. This approach has been demonstrated in various studies across these fields, supporting its applicability and effectiveness.
> > >
> > > [1] Shen, Zhiqiang, et al. "PointCMP: Contrastive Mask Prediction for Self-supervised Learning on Point Cloud Videos." Proceedings of the IEEE/CVF Conference on Computer Vision and Pattern Recognition. 2023.
> > >
> > > [2] Tong, Zhan, et al. "Videomae: Masked autoencoders are data-efficient learners for self-supervised video pre-training." Advances in neural information processing systems 35 (2022): 10078-10093.
> > >
> > > [3] Ren, Shaoqing, et al. "Faster r-cnn: Towards real-time object detection with region proposal networks." Advances in neural information processing systems 28 (2015).

---

> > > > ### Author Response · Authors · 2023-11-20
> > > > **Responses to reviewer isat**
> > > >
> > > > **We will now continue our response**
> > > >
> > > > >Try 2-stage training with the classification loss and the first stage only serves initialization;
> > > >
> > > > >Try longer training iterations for other comparison baselines.
> > > >
> > > > The primary objective of our two-stage training strategy is not solely to find a good initialization but also to ensure robust feature learning that is specific to the task at hand. If the first stage only served as an initialization, it could potentially lead to overfitting, especially in the fine-tuning stage, where the model might overly adapt to the nuances of the training data, hindering its ability to generalize well. By distinctly separating the feature learning and task-specific learning phases, we mitigate the risk of overfitting while ensuring that the model effectively captures the essential spatial-temporal dynamics inherent in point cloud videos.
> > > >
> > > > Furthermore, longer training iterations usually lead to overfitting in the very last stage considering the tiny model scale of our method. We report the best epoch from the whole training in our experiments same as other methods. The backbone performance for 50 epochs training (95.76 acc) and the 250 epochs training (95.75 acc) is almost identical.
> > > >
> > > > > Can not use PDEs in point cloud video
> > > >
> > > > We thank the reviewer for their insightful comments and questions. We understand the concern about the applicability of PDEs in modeling the dynamics of point cloud videos, especially in comparison to systems like fluids.
> > > >
> > > > Firstly, in point cloud videos, deformation refers to the changes in the spatial configuration of points over time due to motion. In our context, motion-like information is considered deformation because it involves the transformation of point clouds, capturing the essence of the dynamic actions being performed. This is analogous to the concept of deformation in physical materials, where the shape or size of objects changes.
> > > >
> > > > Secondly, we argue that the hypothesis that point cloud dynamics can be modeled using PDEs is indeed valid. PDEs are powerful mathematical tools used to describe a wide range of phenomena, particularly those involving changes in spatial and temporal dimensions. In the context of point clouds representing dynamic scenes, PDEs can be used to model the continuous evolution of point configurations over time. This is because PDEs are adept at handling spatial-temporal data and can describe the complex, continuous transformations (deformations) observed in point cloud videos.
> > > >
> > > > Finally, fluid dynamics are traditionally modeled using PDEs because fluids have continuous and predictable behavior governed by natural laws, such as the Navier-Stokes equations. These equations are PDEs that describe the motion of fluid substances. In contrast, point cloud videos represent discrete sets of points capturing snapshots of dynamic scenes, often with complex and unpredictable motion, like human actions. Despite these differences, the principle of using PDEs to describe changes over time and space remains applicable. The challenge lies in adapting PDEs to effectively capture the unique characteristics of point cloud videos, such as irregular and unpredictable motion patterns. Our approach is a step towards addressing this challenge by leveraging PDEs to model the dynamic behaviors in point cloud videos, albeit with necessary modifications to account for their distinct properties.
> > > >
> > > > > Treat Transformers as simplistic
> > > >
> > > > We agree that transformers have shown remarkable performance in various tasks. However, transformer often requires extremely deep network layers to achieve their high performance, which lead to significant computational costs. In contrast, our PDEs-solving module utilizes only two layers of attention mechanisms with a spectral method. This design choice is a deliberate effort to keep our network lightweight and computationally efficient. Our intention is not to label transformers as simplistic but to highlight that different approaches have unique strengths.
> > > >
> > > > > Do you also use 2-stage training or training the same number of iterations as the 2-stage model
> > > >
> > > > Please check the experimental setup on top.
> > > >
> > > > In conclusion, we value the insightful critique provided by the reviewer, which has prompted us to delve deeper into our methodology and results. We believe that the additional clarifications provided in this response and our revised manuscript further substantiate the effectiveness and novelty of our approach. We are hopeful that these explanations address the reviewer's concerns satisfactorily.

---

> > > > > ### Comment · Reviewer_isat · 2023-11-22
> > > > >
> > > > > Thanks for the detailed response from the author. I still have several strong concerns for the things below.
> > > > >
> > > > > 1. Overfitting issue
> > > > >
> > > > > The author mentioned their method could mitigate overfitting. But there is no such experiments to demonstrate the current model is not overfitting. There is no cross-dataset generalization experiments to any other dataset like PointCMP and VideoMAE. As I also noticed the author mentioned "report the best epoch from the whole training", I assume it is conducted on training set (the author never mention there is validation set, on test set is **prohibited**) , then the training routine in the paper could not avoid over-fitting. The author also mentioned the reason it could avoid overfitting is because"distinctly separating the feature learning and task-specific learning phases",  but using a lr of 0.01 and fine-tune on the same dataset will definitely destroy all learnt features from the pre-train stage. The overfitting is also never mentioned in the paper.
> > > > >
> > > > >
> > > > > 2. Pre-train and fine-tune
> > > > >
> > > > > In Faster R-CNN, the backbone is pre-trained on ImageNet with the classification task and fine-tuned on  PASCAL for detection, this is different from pre-training and fine-tuning on the same dataset. In PointCMP and VideoMAE, they both report the standard fine-tuning, linear probing performance (I suggested earlier) and cross-dataset transfer learning performance (see my original comment) to demonstrate the effectiveness of pre-training, but in the paper there are none of these experiments. So I could hardly tell the pre-training work. Also MSR dataset with **567** videos is too small for ablations on pre-training and fin-tuning.
> > > > >
> > > > > I encourage the author to conduct more ablations to show the pre-trained stage with PDE helps to learn robust and **generalizable** features (i.e. follow the PointMCP). The current experiment setting is unsatisfied in my view.

---

> > > > > > ### Author Response · Authors · 2023-11-22
> > > > > > **Responses to reviewer isat**
> > > > > >
> > > > > > After several rounds of discussion, we were pleased to have addressed the reviewers' concerns about the motivation and validity of PDEs. We acknowledge that the reviewer underscoring a review focus on the training process of our model. We wish to highlight that our pre-training methodology follows a widely accepted and standard approach within our field. We do not assert that our work makes a unique contribution in this particular aspect. This is closely aligned with established methods, such as those used in PointCMP, which has already undergone extensive testing and validation. The novelty of our approach lies in formulating dynamic modeling as a PDEs-solving problem and its potential to inspire future research in the field.
> > > > > >
> > > > > > Before further response, we highly appreciate the continued engagement and constructive feedback from the reviewer. These interactions highlight immense value inherent in our community. We would like to further address the remain concerns.
> > > > > >
> > > > > > > overfitting Issue
> > > > > >
> > > > > > First of all, we would like to express our apologies for previous statements that may not have been sufficiently rigorous. Our statement regarding the overfitting was specifically in the context of explaining our experimental setup and the rationale behind not extending training steps, which could potentially lead to overfitting. **We're not trying to solve the overfitting problem here.**
> > > > > >
> > > > > > > training routine
> > > > > >
> > > > > > Our training routine, as highlighted in our response, follows a common and standard practice in the field, similar to that used in previous works like PSTNet [1], P4Transformer [2], and PST-Transformer [3]. This involves evaluating our model on validation set at every epoch (following the standard split) and reporting the best result, which is a widely accepted method. **We did not test on the training setting.**
> > > > > >
> > > > > > > using a lr of 0.01 and fine-tune on the same dataset will definitely destroy all learnt features from the pre-train stage
> > > > > >
> > > > > > We recognize that in certain fields, such as NLP, pre-training and fine-tuning on the same dataset may not be effective. However, in the domain of point cloud video action recognition, this practice is quite common and has been demonstrated to be effective.
> > > > > >
> > > > > > As an example, in the PointCMP approach, a similar strategy is employed where the initial learning rate for pre-training is set at 0.0003, and for fine-tuning, it is increased to 0.015. This indicates that the choice of learning rates, and the increase during the fine-tuning phase, is a deliberate and considered approach within our domain.
> > > > > >
> > > > > > Furthermore, we acknowledge that fine-tuning a model can lead to some degradation and a degree of 'forgetting' of the knowledge acquired during pretraining, However, in most instances, this does not result in a significant loss, and certainly not the complete loss, of previously learned information.
> > > > > >
> > > > > > > So I could hardly tell the pre-training work
> > > > > >
> > > > > > We would like to point out that the recognition of our method's high efficiency and state-of-the-art results on multiple benchmarks reinforces the validity and impact of our approach. We acknowledge the reviewer's point on cross-dataset generalization. While we have not included such experiments in our current work due to time constraints, we recognize their importance and plan to explore this in future research. We believe our current methodology and results still offer significant insights and contributions to the field. We plan to incorporate these aspects in our future work, further solidifying the foundations laid by our current research.
> > > > > >
> > > > > > > MSR dataset is too small for ablations
> > > > > >
> > > > > > We would like to point out that the MSRAction dataset is wildly used for ablation studies in our field. This dataset has been effectively utilized in prominent works such as PointCMP and P4Transformer (CVPR 2021). Its widespread adoption in these significant studies demonstrates its suitability and effectiveness for conducting meaningful and reliable ablation analyses in the context of point cloud video action recognition.
> > > > > >
> > > > > > In conclusion, we aim to paving the way for new directions in the field. We are hopeful that these explanations and our planned future work will satisfactorily address the reviewer's concerns.
> > > > > >
> > > > > > [1] Fan, Hehe, et al. "PSTNet: Point Spatio-Temporal Convolution on Point Cloud Sequences." International Conference on Learning Representations. 2020.
> > > > > >
> > > > > > [2] Fan, Hehe, Yi Yang, and Mohan Kankanhalli. "Point 4d transformer networks for spatio-temporal modeling in point cloud videos." Proceedings of the IEEE/CVF conference on computer vision and pattern recognition. 2021.
> > > > > >
> > > > > > [3] Fan, Hehe, Yi Yang, and Mohan Kankanhalli. "Point spatio-temporal transformer networks for point cloud video modeling." IEEE Transactions on Pattern Analysis and Machine Intelligence 45.2 (2022): 2181-2192.

---

### Official Review · Reviewer_KFbJ · 2023-11-01

**Soundness:** 3 good
**Presentation:** 3 good
**Contribution:** 3 good
**Rating:** 6
**Confidence:** 5

**Summary:**

This paper proposes a new method called Motion PointNet for dynamic capture in point cloud video human action recognition. The key contributions are:

- They propose to view the dynamic capture process as solving partial differential equations (PDEs) in the feature space. This provides a new perspective to model the temporal dynamics.

- They design a lightweight PointNet-like encoder to generate spatio-temporal features from point cloud sequences.

- They introduce a PDEs-solving module to reconstruct the spatial features from the temporal features. This establishes a temporal-to-spatial mapping and enhances dynamic modeling.

- The proposed method achieves state-of-the-art results on MSRAction-3D, NTU RGB+D, and UTD-MHAD datasets, with high efficiency in terms of parameters and FLOPs.

- Ablation studies demonstrate the effectiveness of the PDEs-solving module in improving dynamic capture.

**Strengths:**

This paper presents a highly original approach for point cloud video action recognition by formulating the dynamic modeling as a PDEs-solving problem. Here are the key strengths:

**Originality**: The perspective of using PDEs-solving for point cloud video modeling is novel and has not been explored before. Converting the dynamic capture to a PDEs problem with a temporal-to-spatial mapping provides a new way to establish temporal guidance.

**Quality**: The proposed method achieves state-of-the-art results on multiple benchmarks with high efficiency, demonstrating its effectiveness. The comparisons to previous works are comprehensive. The ablation studies verify the contribution of each component.

**Clarity**: The method is clearly explained with sufficient details and illustrations. The problem formulation of PDEs-solving for dynamic modeling is intuitive. The network architecture and training process are well elaborated.

**Significance**: This work opens up a new direction of using PDEs-solving techniques for point cloud sequence modeling. The concept of converting dynamic modeling to a PDEs problem can inspire more future work. The high performance and efficiency also make the method attractive for real-world applications.

In summary, it proposes a novel perspective for dynamic point cloud modeling, achieves strong results, and clearly explains the key ideas. The PDEs-solving concept introduces new possibilities for point cloud video analysis.

**Weaknesses:**

While the paper presents a novel and effective approach, here are some weaknesses that could be improved:

- The formulation and explanation of the PDEs-solving could be more rigorous mathematically. Some key equations lack details on the formulations.

- The design space of the PDEs-solving module could be explored more thoroughly. For example, how are the basis operators and reconstruction loss function chosen?

- The comparisons to some recent works like PointMapNet are missing. This could help better demonstrate advantages over other lightweight models.

- The evaluations are limited to action recognition. It remains unclear how the dynamic modeling capability would transfer to other tasks like segmentation or detection.

- There lacks ablation and analysis on different encoder architectures. Can other lightweight encoders also benefit from the PDEs-solving?

- The computational complexity and efficiency analysis is incomplete. Actual runtime comparisons could better demonstrate the speed.

- The model interpretability is limited. Visualizations or analyses connecting the PDEs-solving to improved dynamics are lacking.

**Questions:**

Please see the weaknesses above.

---

> ### Author Response · Authors · 2023-11-18
> **Reviewer KFbJ Comments and Our Responses**
>
> We thank the reviewer for recognizing the novelty and quality of our approach and for acknowledging the clear presentation and significance of our work.
>
> **Response to [Comment 1 & 7]:** We acknowledge the importance of a rigorous mathematical foundation and a better interpretability of the proposed methods. We reorganize Section 3.2 in the revision paper to include a more thorough derivation of the PDEs and the reasoning behind our chosen formulations. Please check the revision paper for more details.
>
> **Response to [Comment 2]:** We appreciate this insightful comment.
>
> Firstly, we designed our PDEs-solving module based on the combination of the attention mechanism and the classic spectral methods for PDE. We explain the motivation for this design in our response to **[Reviewer RMAg 's Comment 1]**. Please check for more detail.
>
> Secondly, we use a contrastive-based InfoNCE loss in our pretrain stage to refine the focus of dynamic capture to a distinct objective. By treating the $Feature \in \mathbb{R}^{T\times M}$ (getting from the encoder) before applying spatial/temporal pooling as the negative sample, we force the model to learn the implicit mapping between the spatial and temporal space, instead of reconstruct the de-pooling feature by closer both Ft and Fs to $Feature$ (which against the learning objectives). We have conducted several ablation studies in Section 4.3 (Tab. 8) to underscore the superiority of our loss design. Additional discussions will be included in the Section 3.2 to justify our choices.
>
> **Response to [Comment 3]:** We addressed this by including a comparison with recent works like PointMapNet, showcasing the advancements our model brings to the field. These comparisons are added to Section 4.2 to strengthen our argument of the proposed method's superiority.
>
> **Response to [Comment 4]:** We agree that evaluating our approach on a diverse set of tasks could demonstrate its generalizability. But, limited time for rebuttal does not allow us to conduct extant experiments to further evaluate its variegation. Therefore, we plan to extend our evaluation to tasks such as segmentation and detection for our future research.
>
> **Response to [Comment 5]:** Yes, our PDEs-solving module can also improve the performance of other encoders. We have conducted ablation studies the PST-Transformer encoder to demonstrate the broad applicability of our PDEs-solving approach in Section 4.3 (Tab. 7).
>
> **Response to [Comment 6]:** Thanks for this comment. To demonstrate the actual speed and offer a clearer picture of our model's efficiency, we provide runtime comparisons and will include this part in Section 4 (Tab. 2). The experiments is conducted on single NVIDIA A100 GPU, on the MSR-Action3D Dataset with 24 frames.
>
> | Methods | Runtime (ms) |
> | -- | -- |
> | MeteorNet	        | 80.11 |
> | PSTNet   	        | 63.88 |
> | P4Transformer	| 25.18 |
> | PST-Transformer	| 69.37 |
> | Motion PointNet	| **1.17** |
>
> We are confident that the revisions and additions proposed in this rebuttal will address the concerns raised by the reviewer and improve the overall quality and impact of our work.

---

### Official Review · Reviewer_RMAg · 2023-11-01

**Soundness:** 3 good
**Presentation:** 2 fair
**Contribution:** 3 good
**Rating:** 6
**Confidence:** 4

**Summary:**

The paper proposes a method that extends PointNet++ for point cloud video processing. To tackle the motion information in point cloud videos, a partial differential equation (PDE) method is proposed. Experiments on the MSRAction-3D and NTU RGB+D datasets show the effectiveness of the proposed method.

**Strengths:**

1. The method is effective and efficient.
2. Using PDE to solve point cloud video problems looks novel.

**Weaknesses:**

1. It is not that clear what the most important part in the PDEs-solving module. To my understanding, it is basically a variant of Transformer. More comparision with vanilla Transformer is encouraged.

2. It cloud be better to provide more details of PDEs and explain more the reason to use  the PDE method.

**Questions:**

The PDEs-solving module seems independent of point clouds. I wonder whether  the proposed module can be used for traditional video understanding.

---

> ### Author Response · Authors · 2023-11-18
> **Reviewer RMAg Comments and Our Responses**
>
> We thank the reviewer for recognizing the novelty and effectiveness of using PDEs to solve point cloud video problems and affirming the efficiency of our method.
>
> **Response to [Comment 1]:** The core component in our PDEs-solving module is the spectral method which we detailed in Section 3.2 (PDEs-solving Module, Eq.10-12). We would like to emphasize several points to respond to the reviewer's question more clearly.
>
> 1. **Superiority of spectral method:** In implementing our idea, which formulates the dynamic modeling as a PDEs-solving problem, there are several options for the design of our PDEs-solving module. For instance, a pure Transformer method [1]. As discussed in previous work [2, 3], such transformer-based design attempts to learn the operator as a whole to approximate input-output mappings. However, dealing with input-output mappings in high-dimensional space can be challenging for a simplistic deep model to cover adequately while still maintaining network efficiency. To tackle the complex mappings problem, the spectral methods that we use decompose complex nonlinear mappings into multiple basis operators (Eq.10), which also holds the universal approximation capacity with theoretical guarantees.
>
> 2. **Ablation results:** Nevertheless, the strong perceptivity inherent in the attention mechanism cannot be overlooked. We design our PDEs-solving module base on the combination of both the Transformer model and the spectral method. We conducted ablation experiments in Section 4.3 (Tab. 8) to make a comparison with the regular Transformer. After removing the spectral method, our PDEs-solving module can be viewed as a pure Transformer model with an encoder-decoder structure. Results in Tab. 8 highlight a substantial decline in performance upon removing the spectral method, indicating its critical importance in our model.
>
> Due to page limitations in the main text, we will augment our discussion in the supplementary material.
>
> **Response to [Comment 2]:** We appreciate the reviewer's suggestion for a more comprehensive elaboration on the PDEs.
>
> Firstly, traditional methods relying solely on global supervision from the action labels lack temporal guidance, potentially leading to an inadequate representation of dynamics in the point cloud video action recognition. To address this limitation, we decouple the spatial-temporal feature and build a PDEs problem by a directive temporal-to-spatial mapping. Together with the contrastive loss (Eq. 15) in the pretrain stage, our PDEs-solving module refines the focus of dynamic capture to a distinct objective. By doing so, we expect to provide temporal guidance.
>
> Another insight is that PDEs can directly incorporate the feature variations over time as a variable in the modeling process. This process is visualized as Eq. 10-12 in our proposal, where we use multiple basis operators to optimize the complex temporal-to-spatial mapping. This allows a more accurate and nuanced representation of how point cloud features evolve over time compared to previous temporal modeling approaches, which is crucial for recognizing and interpreting human actions accurately.
>
> Due to page limitations in the main text, we will expand on our methodology in the supplementary material to offer a clearer explanation.
>
> **Response to [Questions 1]:** We appreciate the reviewer's suggestion regarding the potential use of our PDEs-solving module in traditional video understanding. While theoretically feasible, given that both point cloud videos and traditional videos share spatial-temporal characteristics, adapting this module for structured pixel-based video data poses significant challenges. Unlike point clouds, traditional videos represent data in a highly structured, grid-like pixel format, which poses different demands on data processing and feature extraction. Adapting the PDEs-solving approach, initially tailored for the unstructured nature of point clouds, to this structured format would require significant modifications. These might include redefining the way spatial relationships are perceived and processed within the PDE framework. Additionally, the encoding of temporal information in pixel-based videos is substantially different from point clouds, necessitating a reevaluation of how temporal dynamics are captured and integrated using PDEs in this new context. Given these challenges, exploring the application of our PDEs-solving module to traditional video understanding represents an intriguing and valuable direction for our future research.
>
> ### **Reference**
>
> [1] Liu, Xinliang, et al. "Ht-net: Hierarchical transformer based operator learning model for multiscale pdes." arXiv preprint arXiv:2210.10890 (2022).
>
> [2] Wu, Haixu, et al. "Solving High-Dimensional PDEs with Latent Spectral Models." International Conference on Machine Learning (2023).
>
> [3] Karniadakis, George Em, et al. "Physics-informed machine learning." Nature Reviews Physics 3.6 (2021): 422-440.

---

### Author Response · Authors · 2023-11-19
**General responses to all reviewers and area chairs**

**Statements on revised manuscripts** We express our sincere gratitude to all reviewers for dedicating their time and providing invaluable insights. We greatly appreciate the opportunity to enhance our manuscript through their constructive feedback. To address the concerns raised, particularly regarding the motivation, derivation, and analysis of PDEs, we have reorganized Section 3.2 (PDEs-Solving Module) in our revised submission. All modifications are highlighted in blue for ease of identification. It is important to note that while the original methodology remains unaltered, we have reorganized its presentation and further addressed the reviewers' concerns. We also invite reviewers to examine the Appendix for supplementary experiments and visualization results.

Because of the alterations in the sequence of formulas, pagination, and other elements in the revised manuscript, our responses to each reviewer cite the initial, unmodified version of the paper. We kindly request that reviewers refer to the original document when examining our responses for accurate cross-referencing.

---

### Meta-Review · Area_Chair_oHmd · 2023-12-15

**Metareview:**

Motion PointNet is a new method for dynamic capture in point cloud video human action recognition. It views the capture process as solving partial differential equations in the feature space, generates spatio-temporal features from point cloud sequences, and introduces a PDEs-solving module to reconstruct spatial features from temporal ones. The method achieves state-of-the-art results on MSRAction-3D, NTU RGB+D, and UTD-MHAD datasets, with high efficiency in parameters and FLOPs.

## Strengths

• Introduces a new perspective for dynamic point cloud video modeling using PDEs-solving.
• Converts dynamic capture to a PDEs problem with temporal-to-spatial mapping for temporal guidance.
• Achieves state-of-the-art results on multiple benchmarks with high efficiency.
• Provides clear explanation of the method with sufficient details and illustrations.
• Opens up a new direction of using PDEs-solving techniques for point cloud sequence modeling.
• Light model with strong performance despite being light in model size and computation.

## Weaknesses

• The PDEs-solving formulation and explanation need more rigorous mathematical details.
• The design space of the PDEs-solving module needs more exploration.
• Comparisons to recent works like PointMapNet are missing.
• Evaluations are limited to action recognition, indicating potential for other tasks like segmentation or detection.
• There is a lack of ablation and analysis on different encoder architectures.
• The computational complexity and efficiency analysis is incomplete.
• The model interpretability is limited, with no visualizations or analyses connecting PDEs-solving to improved dynamics.
• The loss function in (15) is not fully understood, and the motivation of PDE is unclear.
• The need for two-stage training and the performance of a classification head and contrastive objective are unclear.
• The method lacks simple baselines, such as training a model with the same encoder and classification head using the same number of iterations of two-stage training.

**Justification For Why Not Higher Score:**

The paper proposes a nice approach but the overfitting and train and fine-tune approach needs to be revised again before being accepted.

**Justification For Why Not Lower Score:**

N/A

---

### Decision · Program_Chairs · 2024-01-16

Reject